# The Crosstalk between FAK and Wnt Signaling Pathways in Cancer and Its Therapeutic Implication

**DOI:** 10.3390/ijms21239107

**Published:** 2020-11-30

**Authors:** Janine Wörthmüller, Curzio Rüegg

**Affiliations:** Laboratory of Experimental and Translational Oncology, Pathology, Department of Oncology, Microbiology and Immunology (OMI), Faculty of Science and Medicine, University of Fribourg, CH-1700 Fribourg, Switzerland

**Keywords:** FAK, Wnt, cell signaling, cancer, malignant mesothelioma, clinical trials, combinatorial therapy

## Abstract

Focal adhesion kinase (FAK) and Wnt signaling pathways are important contributors to tumorigenesis in several cancers. While most results come from studies investigating these pathways individually, there is increasing evidence of a functional crosstalk between both signaling pathways during development and tumor progression. A number of FAK–Wnt interactions are described, suggesting an intricate, context-specific, and cell type-dependent relationship. During development for instance, FAK acts mainly upstream of Wnt signaling; and although in intestinal homeostasis and mucosal regeneration Wnt seems to function upstream of FAK signaling, FAK activates the Wnt/β-catenin signaling pathway during APC-driven intestinal tumorigenesis. In breast, lung, and pancreatic cancers, FAK is reported to modulate the Wnt signaling pathway, while in prostate cancer, FAK is downstream of Wnt. In malignant mesothelioma, FAK and Wnt show an antagonistic relationship: Inhibiting FAK signaling activates the Wnt pathway and vice versa. As the identification of effective Wnt inhibitors to translate in the clinical setting remains an outstanding challenge, further understanding of the functional interaction between Wnt and FAK could reveal new therapeutic opportunities and approaches greatly needed in clinical oncology. In this review, we summarize some of the most relevant interactions between FAK and Wnt in different cancers, address the current landscape of Wnt- and FAK-targeted therapies in different clinical trials, and discuss the rationale for targeting the FAK–Wnt crosstalk, along with the possible translational implications.

## 1. Introduction

Focal adhesion kinase (FAK) is a key regulator of growth factor receptor and integrin-mediated signaling that coordinates multiple fundamental processes such as migration, invasion, angiogenesis, cell survival, or epithelial–mesenchymal transition (EMT), in both normal and cancer cells. Due to this prominent role in cancer, a vast number of preclinical and clinical studies have explored different strategies to block FAK signaling for therapeutic purposes. Molecular and cellular mechanisms of FAK action on both tumor and stromal cell biology, regulation of its activity, and FAK-targeted anticancer therapeutic strategies that are, nowadays, under preclinical and clinical development have been extensively reviewed in different excellent review articles [1,2,3,4]. However, the complex crosstalk and regulatory loops modulated by FAK, in particular concerning the Wnt signaling pathway, have not been explored and reviewed in detail.

Wnt signaling it is one of the most fundamental and conserved regulatory systems in animals. This pathway controls many critical aspects of development and adult tissue homeostasis by regulating cell proliferation, differentiation, migration, genetic stability, and apoptosis, as well as by maintaining adult stem cells in a pluripotent state. Dysregulation of this pathway is associated with a broad spectrum of human diseases, ranging from neurodegenerative disorders to fibrosis, osteoporosis, and cancer. Consequently, Wnt-related research has become a major area for targeted drug discovery and therapeutic development [5,6]. For a long time, the Wnt pathway was considered undruggable, due its complexity and multiple sub-branches. However, more recently, new emerging insights into the pathway mechanisms and crosstalk with other major pathways has led to the development of multiple inhibitory compounds targeting this pathway at different levels. Several Wnt-signaling inhibitors are under early clinical testing with promising outcomes; however, no Wnt-specific drugs have been approved for clinical usage so far [7].

A crosstalk between pathways occurs when the combinatorial signal from both pathways produce a different response than the one triggered by each individual pathway [8]. It must be considered, however, that perturbations to a signaling network, such as the inhibition of a small molecule or pathogenic mutations in one pathway, may affect different tissues or cell types in different ways, highlighting the importance of the tissue-specific context [9]. Feedback loops and crosstalk between signaling pathways can significantly impact the efficacy of cancer therapeutics and cause resistance to targeted agents, thereby representing a major barrier to effective treatments [10]. Understanding the existing interactions between pathways represents a major challenge but may ultimately contribute to identify new, yet unexplored, therapeutic options. In this review we summarize recent findings and elaborate general considerations of the FAK and Wnt signaling pathways and current insights of the crosstalk existing between both pathways with emphasis on their impact on development and most importantly in cancer. We also provide an overview of current strategies and clinical trials that aim to antagonize FAK and Wnt signaling in cancer and the major challenges that are associated with such approaches. Furthermore, we analyze the potential therapeutic benefits of a combinatorial therapy targeting the crosstalk FAK–Wnt.

## 2. The Focal Adhesion Kinase (FAK) Signaling Pathway

FAK is a non-receptor protein tyrosine kinase that mediates growth factor- and adhesion-dependent signaling through several downstream pathways leading to cell migration, invasion, cell cycle progression, and survival [11,12,13], during both development and malignancy. The human gene encoding FAK, named *PTK2*, is located on chromosome 8q24.3 [14] and is highly conserved with over 90% amino acid sequence identity across different species, including human, mouse, chicken, and Xenopus [15,16,17,18]. Its structure is composed of four major domains: (1) a central kinase domain, (2) flanked by an N-terminal four-point-one ezrin–radixin–moesin (FERM) domain, (3) proline-rich regions, and (4) a focal adhesion targeting (FAT) C-terminal domain [3]. The FAK FERM domain represents a critical element in FAK activation. It has a scaffolding role, mediating protein–protein and protein–lipid interactions, and regulates the subcellular localization of FAK [2]. The kinase domain of FAK is localized in the middle of the molecule and contains the activation loop and important tyrosine (Y) residues, such as Y397, Y576, and Y577. FAK tyrosine Y397 serves as the major site of autophosphorylation [3,19], as well as the binding site for various interacting partners, including Src family kinases (SFKs) and the p85 subunit of phosphatidylinositol 3 kinase (PI3-K) [20,21]. Activated FAK–Src promotes cell motility, cell cycle progression, and cell survival, leading to tumor growth and metastasis [22]. For instance, Src–FAK signaling promotes E-cadherin internalization during cancer progression, thus facilitating EMT and enhancing tumor cell motility [23].

Although FAK itself has not been demonstrated to be an oncogene, FAK overexpression and activation have been reported in tumors of broad tissue origin [12], especially in invasive and metastatic tumors [24], including thyroid [25], prostate [26], colorectal [27], and ovary [28] cancers, as well as in malignant mesothelioma [29]. In addition, several studies have associated FAK overexpression with poorer clinical prognosis [30]. However, the exact molecular mechanisms responsible for the increase in FAK expression are not fully deciphered. One of the proposed mechanisms of FAK overexpression is via FAK amplification; although increased FAK expression also occurs independently of FAK gene amplification [31], indicating that transcriptional and/or post-transcriptional mechanisms may also play a role.

FAK signaling is activated by several different mechanisms that include interaction with integrins, growth factor receptors, G protein–coupled receptors, or cytokine receptors [32,33]. The most frequently described mechanism involves engagement of integrins with the extracellular matrix and the subsequent association of proteins, such as talin and paxillin, with the cytoplasmic tail of integrin β subunits [34]. This leads to the recruitment of FAK to sites of integrin clustering via interactions with integrin-associated proteins, leading to FAK activation [3]. Its activity is mediated through kinase-dependent and kinase-independent (scaffolding) mechanisms [1]. FAK kinase-dependent functions are often associated with integrin-related signaling at focal adhesions, where FAK plays an important role in cellular migration and adhesion in both normal and cancer cells [35,36]. FAK also functions as a scaffold protein and participates in protein–protein interactions through its kinase-independent functions. Interestingly, FAK is also able to translocate to the nucleus, where it interacts directly with p53 to promote cell proliferation and survival through p53 degradation [11,37] in a kinase-independent manner. In addition, nuclear FAK has been described to control other transcriptional networks such as the inflammatory signaling pathway, immune escape, and angiogenesis; however, the mechanisms regulating FAK in the nucleus remains unclear [38].

## 3. The Wnt Signaling Pathway

The Wnt signaling pathway is an ancient and highly conserved pathway that regulates crucial aspects of cell fate determination, cell migration, cell polarity, neural patterning, and organogenesis during embryonic development [39]. It has also determinant roles in tissue regeneration and adult homeostasis. Given its importance it is not surprising that mutations in the Wnt pathway are frequently observed in cancer, most notably in tissues that normally depend on Wnt for high self-renewal or repair [40].

The Wnt family consist of 19 secreted glycoproteins that transduce signals by binding to frizzled (Fzd) receptor complexes [41]. Based on the dependence of its key mediator β-catenin, the pathway is in turn subdivided in canonical (β-catenin-dependent) and non-canonical (β-catenin-independent) signaling pathways [42]. The canonical Wnt signaling pathway is driven by β-catenin, a scaffold protein, linking the cytoplasmic tail of classical cadherins via α-catenin to the actin cytoskeleton. In the absence of Wnt stimulation, cytoplasmic β-catenin levels are maintained low by proteasome-mediated degradation controlled by a multiprotein complex consisting of Axin, Adenomatous polyposis coli (APC), Casein kinase1a (CK1a), and Glycogen synthase kinase 3-β (GSK3-β) that bind to β-catenin [43]. The non-canonical Wnt pathways include the planar cell polarity (PCP) and the Wnt/Ca^2+^ pathways. In the Wnt/Ca^2+^ pathway, Wnt-Fzd binding activates phospholipase C (PLC) via G proteins, leading to diacylglycerol (DAG) production and an increase of intracellular Ca^2+^ levels, subsequently activating downstream effectors that include protein kinase C (PKC) [44]. The increased intracellular Ca^2+^ levels also activate calcineurin and calmodulin-dependent protein kinase II (CaMKII) [45]. The PCP pathway, also referred as the Wnt/Jun-N-terminal kinase (JNK) pathway, activates small GTPases, including RhoA, Rac, Cdc42, and JNK [46]. This pathway controls cell polarity, cytoskeletal remodeling, directional cell migration and c-Jun-dependent transcription. As these two pathways are involved in cytoskeletal changes and cell migration, their activation has been associated with cancer cell invasion and metastasis [47,48]. In addition to mutational activation, the Wnt pathway can be aberrantly activated by overexpression of its components, such as Wnts or their Fzd receptors [49].

## 4. FAK–Wnt Pathways Crosstalk in Development

FAK is crucial during development, tissue regeneration and wound healing. Deletion of FAK during mouse development results in lethality at embryonic day 8.5 and a block in cell proliferation. FAK activity is also essential for developmental processes controlling blood vessel formation, cell motility and polarity [32]. FAK-deficient embryos display a general defect of mesoderm development and cells from these embryos have a reduced mobility in vitro [50], while FAK-null embryos present impairment in angiogenesis and defects in the proper development of the heart and blood vessels [51]. In neurons, FAK functions downstream of netrins to promote neurite outgrowth and axon guidance [52]. Several studies have also described FAK playing a role in the development of the placenta, as well as in the musculoskeletal, genitourinary, and respiratory organ systems [53].

Beyond development, Wnt signaling exerts crucial roles in adult tissues and organs, in daily processes of tissue homeostasis and regeneration, in organ repair after injury, and in adult neurogenesis [54]. The best-characterized function of Wnt signaling in adult tissues is the maintenance of stem cell homeostasis, where Wnt ligands promote proliferation and self-renewal of tissue-specific stem cells [55] in the intestine, stomach, skin, liver, and the mammary gland [54].

There are many described interactions between FAK and Wnt during embryonic development (reviewed in [56]). FAK and Wnt are involved in controlling apical cell morphogenesis in *Drosophila* ovarian morphogenesis [57] and in regulating early patterning in the nervous system of *Xenopus laevis* [58], where FAK regulates Wnt3a gene expression to control cell fate specification in the developing neural plate. Both pathways have been also shown to be implicated in bone remodeling; FAK promotes osteoblast progenitor cell proliferation and differentiation by enhancing Wnt signaling [59]. In addition, FAK was shown to play a pivotal role in promoting BMP9-induced osteogenesis of synovial mesenchymal stem cells via the activation of Wnt and MAPK pathways [60], while another study demonstrated that FAK and BMP-9 synergistically trigger osteogenic differentiation and bone formation of adipose tissue-derived stem cells by enhancing Wnt-β-catenin signaling [61]. FAK and Wnt signaling are also involved in maintaining normal intestinal homeostasis and promoting mucosal regeneration following DNA damage, with FAK required downstream of Wnt signaling for Akt/mTOR activation [62]. More recently it was found that both, the Stat3 pathway and Wnt signaling cooperatively regulate the survival of the epithelial cells in the damaged mucosa and isolated crypts through activation of integrin/FAK signaling [63]. FAK also plays a role in the control of the epidermal stem cells via a mechanism that involves crosstalk with the Wnt/β-catenin pathway [64].

## 5. FAK–Wnt Pathways Crosstalk in Cancer

Given Wnt’s essential role in embryonic development, tissue homoeostasis, and stem cell biology, this pathway must be tightly regulated; its dysregulation has been associated with many types of cancer. “No man is an island”, and similarly no pathway is modulated without affecting others [5]. Understanding how FAK regulates Wnt transcription and pathway activation during development, and more importantly, during cancer progression, could offer new potential opportunities for cancer treatment [56].

### 5.1. Colorectal and Intestinal Cancers

Colorectal cancer (CRC) is the second leading cause of cancer morbidity and mortality worldwide [65]. Genetic alterations in Wnt signaling occur in over 90% of human sporadic CRC, among which inactivation of the tumor suppressor adenomatous polyposis coli (*APC*) occur in 85% of the cases, while activating mutations of the proto-oncogene β-catenin (*CTNNB1*) occur only in 5% of the cases [66]. Activating mutations of BRAF (mostly V600E mutation), a serine/threonine kinase downstream of KRAS and upstream of MEK (mitogen-activated protein kinase kinase) have been reported to occur in approximately 10% of all CRC patients [67]. While Wnt signaling has been shown to function as tumor suppressor in other cancers [68], activation of this pathway seems a critical step in the tumor initiation and development of CRC [69].

In the absence of Wnt, the canonical function of APC is to form a “destruction complex” with Axin/Axin2, CK1-α, and GSK-3β promoting the sequestration, ubiquitination, and subsequent proteasomal degradation of β-catenin [70]. APC is an essential component of the cytoplasmic protein complex that targets β-catenin for destruction [71]. The loss of functional APC results in less efficient GSK3 (GSK3α and GSK3β)-mediated phosphorylation of β-catenin, leading to reduced degradation of β-catenin, thus mimicking Wnt stimulation. β-catenin then accumulates and enters the nucleus to interact with the transcription factors TCF (T-cell factor)/LEF (lymphoid enhancing factor) to activate the transcription of Wnt target genes [72], including cyclin D1 and c-myc, among others. The tumorigenic consequences of dysregulated β-catenin activity are the stimulation of cellular growth and proliferation, and the disruption of differentiation programs [70].

As mentioned above, GSK3β is a central player in the canonical pathway; it operates by regulating the phosphorylation and degradation of β-catenin, and its activity is tightly controlled [73]. Gao et al. recently reported a novel FAK–Wnt regulation axis: FAK and PYK2 were found to be elevated in adenomas in APC^min/+^ mice and in human CRC tissues, in addition they promoted Wnt/β-catenin pathway activation by phosphorylating GSK3β leading to β-catenin accumulation, which in turn initiates and supports intestinal tumorigenesis. Pharmacological inhibition of FAK/PYK2 repressed adenoma formation in APC^min/+^ mice and reduced intestinal levels of phospho-GSK3β and β-catenin, indicating that the FAK/PYK2/GSK3β axis is critical in APC-driven intestinal tumorigenesis [74]. Altogether this suggests that drugs inhibiting simultaneously FAK and PYK2 may be an effective treatment for CRC. In line with this, recent studies showed that BRAF inhibitors upregulate the Wnt/β-catenin pathway in BRAF^V600E^-mutant CRC cell lines through FAK activation-mediated phosphorylation of GSK3. Co-targeting of BRAF/Wnt pathways or BRAF/FAK pathways exerted strong synergistic antitumor effects in cell culture and mouse xenograft models [75]. Given that Wnt/β-catenin pathway activation is one of the causes of CRC resistance to BRAF inhibition, co-targeting the Wnt/β-catenin pathway using FAK inhibitors may represent a new feasible solution to overcome resistance to BRAF inhibitors with great potential to change the landscape for patients with *BRAF* mutations [76].

Interestingly, FAK inhibition with the small molecule inhibitor Y15 increased DKK1, a known inhibitor of the Wnt pathway that plays an important role in CSC regulation in the metastatic CRC cell line, SW620. Y15 also downregulated Wnt pathway genes, such as *LRP5* and *FZD2*, and upregulated *SFRP5*, one of several members of the SFRP family that controls Wnt pathway signaling, specifically by preventing it from binding to its receptor [77].

Another link between Wnt and FAK signaling is the finding that the expression of HEF1 (human enhancer of filamentation 1), implicated in progression of CRC, was also shown to modulate both pathways. HEF1 localizes to focal adhesions to coordinate FAK and Src signaling cascades in integrin-dependent adhesion, migration, invasion, and survival [78,79]. In a study evaluating the role of HEF1 in CRC tumorigenesis, HEF1 was identified as a novel Wnt signaling target and classified as a biomarker for tumor aggressiveness; overexpression of HEF1 in CRC cells enhanced cell migration and invasion. In addition, the expression of HEF1 was upregulated by Wnt-3a, β-catenin, and Dvl2 in a dose-dependent manner, and it was suppressed following β-catenin downregulation by sh*RNA* [80].

In conclusion, there is an unequivocal evidence that FAK and Wnt pathways play a role in regulating CRC initiation and progression. These findings suggest that pharmacological inhibition of FAK might be effective in the treatment of CRC [81,82].

### 5.2. Malignant Mesothelioma and Lung Cancer

An interesting correlation between FAK and Wnt signaling was found in malignant mesothelioma (MM), an aggressive neoplasm that develops from the mesothelial cells lining the pleural, peritoneal, and pericardial cavities [83]. Treatment with the a FAK inhibitor in different MM cell lines strongly activated the Wnt signaling pathway; more specifically, it increased p-JNK T183/Y185 and total JNK levels. Conversely, Wnt inhibition in the same cells led to FAK activation, increasing p-FAK Y397 and total FAK levels; indicating an antagonistic regulation of these two pathways [84]. Simultaneously blocking FAK and Wnt signaling drastically reduced cell proliferation and survival of MM cell lines. Both pathways were already described to independently play a role in MM by promoting different tumorigenic properties; dysregulated Wnt signaling was implicated in invasion and resistance to apoptosis [85,86], while FAK signaling was shown to promote invasion and EMT [29].

A relation between FAK and Wnt signaling was also found in a study evaluating the function and mechanism of FAK in regulating the inflammatory response in the A549 cell line, a model for non-small cell lung cancer (NSCLC). The inhibition of FAK decreased the activation of the Wnt and NF-κB signaling pathways, accompanied by a reduction in inflammatory activity [87]. In another study using the same cell line, FAK was shown to act upstream the Wnt/β-catenin pathway. When A549 cells were treated with Maclurin, a natural organic compound which can be extracted from *Morus alba*, the c-Src/FAK and ERK pathways were inactivated through the reduction of ROS levels. This in turn activated GSK3β, leading to the inactivation of the β-catenin transcription factor. As a result, migration and invasion of A549 cells were attenuated, demonstrating that the anti-metastatic effect of Maclurin was exerted by antioxidative activity and inhibition of Src/FAK–ERK–β-catenin signaling pathway [88]. Similarly, another study described FAK as an upstream regulator of Wnt signaling, as it modulated cell proliferation and migration through the activation of GSK3β and subsequent decrease of β-catenin and MMPs in the human lung adenocarcinoma CL1 cells after treatment with the leaf extract of *Momordica charantia* [89].

Both Wnt/β-catenin and FAK pathways were independently found to be implicated in idiopathic pulmonary fibrosis (IPF). Blocking Wnt signaling strongly attenuated and reversed fibrosis in a murine IPF model system [90]. Similarly, FAK inhibition attenuated fibrotic responses to lung injury in vivo [91]. Whether FAK and Wnt signaling pathways are somehow connected in regulating IPF disease progression needs further evaluation [56].

### 5.3. Ovarian and Breast Cancer

In a model of high-grade serous ovarian cancer (HGSOC) it was recently shown that FAK activity sustained intrinsic and acquired cisplatin resistance via a Wnt/β-catenin-Myc ‘stemness’ pathway [92]. Previous studies have already linked increased activity of FAK with chemoresistance in ovarian carcinoma cells [93]. More recently beta integrin-related protein 1 (ITGBL1), an integrin-like member of the EGF-like protein family rich in cysteine-rich repeats, was found highly expressed in ovarian cancer tissues, promoting ovarian cancer cell migration and adhesion through Wnt/PCP and FAK/Src pathways in vitro [94]. However, the specific mechanisms on how ITGBL1 affects both pathways or if there is a synergistic effect is still not clear and will need further investigation. Different studies support the role of ITGBL1 in the regulation of cancer progression. ITGBL1 was reported to promote breast cancer bone metastasis through TGF-β signaling [95] (being FAK a known mediator of TGF-β signaling [96]) and CRC migration and invasion, possibly also through the FAK signaling pathway [97]. In another study, beta 1 integrin (ITGB1) was identified as a key mediator for Twist-induced EMT, where Twist was shown to induce EMT and cell motility in breast cancer, by regulating a complex signaling network consisting of ITGB1-FAK/ILK axis, PI3K/AKT, MAPK/ERK, Wnt, and P53 signaling [98]. ITGB1 is a component of beta 1 integrin heterodimers, a family of major adhesion receptors for ECM proteins, such as fibronectins, collagens, and laminins, and is a key activator for FAK and ILK pathways [99]. In this context, the activation of ITGB1-FAK/ILK axis seems essential for stimulating Wnt signaling [98]. Similar to CRC, the *APC* tumor suppressor gene also plays a significant role in breast cancer carcinogenesis. APC is silenced by hypermethylation or mutated in about 70% of human breast cancers [100]. Recently it was reported that APC mutation promoted mammary tumor cell proliferation in a PyMT mouse model through the activation of Src and JNK signaling downstream of FAK [101]. In addition, inhibiting Src or JNK diminished the APC-mediated cell proliferation, suggesting that targeted inhibition of these signaling pathways downstream FAK appears as a promising strategy in APC-mutated breast cancers. In a study on ductal carcinoma in situ (DCIS), Williams et al. showed that CSC-enriched populations were more radio-resistant due to higher levels of FAK activity, and inhibition of FAK both in vitro and in vivo decreased the self-renewal capacity of these cells, accompanied by a reduction in Wnt3a and β-catenin levels. Wnt signaling stimulation after FAK inhibition rescued the self-renewal capacity, indicating once more that Wnt functions downstream of FAK signaling [102]. Similarly, another study showed that pharmacological inhibition of FAK in mouse xenograft models of triple-negative breast cancer (TNBC) reduced the proportion of CSCs in the tumors. In addition, FAK inhibition blocked β-catenin activation and the introduction of an active mutant form of β-catenin reversed the preferential targeting of CSCs by FAK inhibition [103].

### 5.4. Other Malignancies

While in some cancer types the relationship between FAK and Wnt is already well evident, in other cancers this crosstalk is still elusive. Different studies of hepatocellular carcinoma, melanoma and acute myeloid leukemia suggest that FAK operates upstream of Wnt, while in prostate cancer Wnt seems to acts upstream of FAK. In addition, a study of renal carcinoma reveals that Wnt and FAK might act concomitantly in promoting cancer progression.

Deletion of FAK in hepatocytes blocks tumor proliferation and prolongs the survival in a c-Met/β-catenin-driven hepatocellular carcinoma (HCC) mouse model [104], suggesting that FAK is required for c-Met/β-catenin-driven hepatocarcinogenesis. More recently it was demonstrated that overexpressed FAK and β-catenin cooperate to induce HCC in mice [105] and that FAK functionally stimulates Wnt/β-catenin signaling, activates CSC traits, and drives tumorigenicity in HCC cells [106]. The FAK gene is particularly overexpressed in human malignant melanoma cells, leading to constitutive high levels of pVE-cadherin (Y658). This elevated pVE-cadherin (Y658) allows for upregulation of Kaiso-dependent genes in the nucleus (*WNT11* and *CCDN1*), accelerating the vasculogenic mimicry capacity in these cells, which is associated with a high tumor grade, invasion, metastasis, and a short survival [107]. A study on acute myeloid leukemia (AML) also supports a crosstalk between FAK and Wnt signaling. In this case, FAK splice variants are abnormally expressed, and this dysregulation seems to play a key role in maintaining primitive AML cells, by altering Wnt signaling and β-catenin activity [108]. In a study of human renal cell carcinoma (RCC), the proto-oncogene Golgi phosphoprotein 3 (*GOLPH3*) was able to activate simultaneously the FAK/Raf/MEK pathway and Wnt/β-catenin promoting RCC cell proliferation and malignancy. However, in this scenario, whether FAK and Wnt act independently but concomitantly or interdependently from each other was not addressed [109]. A more intricate interaction between Wnt and FAK was found in a study of prostate cancer where Wnt-induced secreted protein-1 (WISP-1) promoted migration in human prostate cancer cells by downregulating miR-126 expression via αvβ1 integrin, FAK, and p38 signaling pathways [110]. Another study demonstrated that silencing Cripto-1 (CR-1) expression in prostate cancer cells, suppressed proliferation, migration, and invasion of these cells through the inhibition of both FAK/Src/PI3K and Wnt/β-catenin signaling pathways [111].

## 6. Current Landscape of Wnt- and FAK-Targeted Therapies in Clinical Trials

Despite the increasing number of new targeted anticancer agents available in the clinic, treatment resistance and failure remain major challenges in the management of most advanced solid cancers, including breast [112] and colorectal cancers [113], and malignant mesothelioma [114]. There are multiple compensatory mechanisms that counterbalance the therapeutic effects of targeted-drugs, ultimately leading to treatment failure [115], such as the inactivation of apoptotic factors or enhancement of cell survival pathways [116]. Among promising strategies to improve clinical outcomes are the use of combinatorial therapy to target the distinct adaptive response mechanisms or the disruption of crosstalks existing between different signaling pathways, i.e., targeting downstream proteins or transcriptional factors to reduce the resistance or improve the effects of targeted therapies [117].

In addition to the roles of Wnt signaling in development, dysregulated Wnt also contributes to cell proliferation, chemoresistance and enhanced tumorigenic potential of CSC, all of which are factors involved in tumor recurrence after therapy, resistance to further anticancer therapies, and poor survival [118]. Therefore, targeting Wnt signaling seems theoretically beneficial at multiple levels: inhibition of tumor growth and survival with minimal effects on somatic cells, inhibition of CSC maintenance (and consequently tumor relapse), and prevention (or reversal) of tumor resistance [119]. However, the complexity of this pathway with its multiple sub-branches and the essential role of Wnt in regulating adult homeostasis or bone remodeling underlie the practical difficulties of finding therapeutically relevant and “safe” Wnt-targeting agents. In this section, we describe some of the Wnt and FAK inhibitors that have reached or are currently being tested in early phase clinical trials.

In general, agents targeting Wnt signaling include compounds at the ligand/receptor level (Vantictumab, Ipafricept, Rosmantuzumab, Foxy-5, and OTSA101-DTPA), transcriptional level (CWP232291, PRI-724, and SM08502) and at the level of Wnt secretion (Porcupine inhibitors: WNT974 and ETC-159, RXC004, and CGX1321) (Table 1) (for more details see additional reviews [7,119,120]). In the specific cases of Vantictumab (OMP-18R5; a monoclonal antibody that binds to Fzd receptors [121]) and Ipafricept (OMP-54F28, a recombinant fusion protein that acts as a soluble decoy receptor sequestering secreted Wnts [122]), several clinical trials have evaluated their effects as single agents or in combination with Paclitaxel, Carboplatin, or Gemcitabine in advanced solid tumors. Although both Vantictumab and Ipafricept were in principle well tolerated, they ultimately caused important bone metabolism disorders, i.e., an increased incidence of fractures in patients, therefore restricting the future use of these compounds [123,124]. Given the important role Wnts play in the differentiation of osteoblasts and osteoclasts, it is not surprising that Wnt inhibition affects bone homeostasis [125]. In the case of Rosmantuzumab (OMP-131R10), an anti-R-spondin 3 antibody, there is little information about the outcome of the phase I clinical trial (clinical trial identifier: NCT02482441), being just reported that Rosmantuzumab “failed to provide compelling evidence of clinical benefit” [126]. Regarding Foxy-5, a synthetic Wnt5a-mimicking peptide [127], after having completed a phase I study this compound is currently being tested as neo-adjuvant therapy in a phase II clinical trial aimed at evaluating the anti-metastatic activity of Foxy-5 in subjects with resectable colon cancer (NCT03883802). OTSA101-DTPA, a radiolabeled monoclonal antibody against Fzd10, is currently in phase I testing to assess its safety and pharmacokinetics in patients with relapsed or refractory synovial sarcoma (NCT04176016).

Inhibitors at the transcriptional level include PRI-724, CWP232291 and SM08502. While the small molecule PRI-724 inhibits the recruiting of β-catenin with its coactivator CBP, the small-molecule CWP232291 (CWP291) binds to Sam68, decreasing the expression of β-catenin target genes. In a phase I study, CWP291 was considered safe and demonstrated single-agent activity in acute myeloid leukemia patients, with plans for future combination studies [128]. Regarding PRI-724, despite some adverse events described in different phase I clinical trials that evaluated its effect in advances solid tumors including pancreatic cancer, and chronic myeloid leukemia, this compound was considered overall safe with “modest clinical activity”. Although it seems that it will continue being tested in the setting of fibrosis, no follow-up is foreseen for cancer therapy at the moment [119]. The small molecule SM08502 was shown to reduce Wnt pathway signaling and gene expression through the potent inhibition of CDC-like kinase (CLK) activity [129] and is currently in “recruiting” state in a phase I clinical trial for patients with advanced solid tumors (NCT03355066).

One of the emerging and most promising ways for targeting Wnt signaling is to block Wnt ligand production through the inhibition of the acyl-transferase Porcupine [130]. Wnts need to be coupled to fatty acids to be secreted, this occurs in the Endoplasmic Reticulum by Porcupine. Acylation of Wnts allows binding to Wntless (Wls) in the Golgi apparatus, which in turn facilitates secretion of the mature Wnts [131]. There are currently several competing inhibitors of Porcupine being evaluated in different clinical trials. Wnt974 (LGK974) and ETC-159 are under phase I studies in “recruiting” state (NCT01351103 and NCT02521844, respectively). However, similar to other Wnt inhibitors, skeletal side effects such as impairment of bone mass and increase in bone resorption are described, supporting the notion of adding bone protective agents when treating patients with these type of inhibitors [125]. RXC004 and CGX1321, other potent selective orally bioavailable inhibitors of Porcupine, are also in “recruiting” state in phase I studies in patients with advanced solid tumors (NCT03447470 and NCT02675946, respectively).

Several other small molecules antagonizing the interaction between β-catenin and TCF/LEF1, such as PNU74654, 2,4 diamino-quinazoline, PKF115-584, or CGP 049090, have been identified. Other small molecules include those targeting Dvl (NSC 668036, FJ9, 3289-8625), CK1, AXIN, or Tankyrase 1 and 2 (that promote AXIN stabilization). Natural compounds, such as vitamin D, curcumin, quercetin, or resveratrol, or existing drugs in the market such as Sulindac and Celecoxib are also being investigated in preclinical studies for Wnt signaling inhibition [132].

Taken together, the data and rationale behind using Wnt inhibitors supports them as promising therapeutics. Nevertheless, despite extensive research and some promising outcomes from preclinical and clinical data, enthusiasm about Wnt inhibitors needs to be carefully evaluated. Firstly, the intricate nature of Wnt signaling and crosstalk with other signaling pathways are not fully elucidated. This lack of understanding most probably hinders the development of more efficacious therapeutic strategies. Secondly, we do not know whether the inhibition of Wnt signaling will activate resistance response mechanisms by other signaling pathways. Thirdly, the important role of Wnt signaling in physiological homeostasis and in bone remodeling, and the consequent adverse effects of its inhibition, cannot be dismissed. It is noteworthy to mention that toxicities originated from these treatments may vary dramatically depending on the targeted pathway component [132].

In the case of FAK inhibitors, translation in the clinical setting is more advanced: four adenosine triphosphate (ATP)-competitive, small-molecule inhibitors of FAK (GSK2256098, BI 853520, PF-00562271, and VS-6063), have been proven effective in several preclinical studies [133] and have advanced into phase I and II clinical trials (Table 2). Currently two (GSK2256098 and VS-6063) are in ongoing clinical trials evaluating its use in combination with other chemotherapeutic agents and/or cell signaling/checkpoint inhibitors in order to increase their efficacy.

In a phase I study (NCT01138033), GSK2256098 showed clinical activity in patients with malignant mesothelioma, particularly those with merlin (encoded by *NF2*) loss [134]. With the premise that combined FAK and MEK inhibition may provide greater anticancer effect than FAK monotherapy, another phase I study evaluated GSK2256098 in combination with Trametinib (NCT01938443). Mesothelioma patients with loss of merlin had longer progression-free survival than subjects with wild-type *NF2*, although support for efficacy with this combination was limited [135]. Recently, another study is attempting to evaluate this combination in patients with advanced pancreatic cancer (NCT02428270).

Two studies have evaluated the effect of BI 853520 monotherapy in advanced or metastatic tumors (NCT01335269 and NCT01905111). These studies have demonstrated that BI 853520 has a manageable and acceptable safety profile and preliminary antitumor activity. However, as single-agent its activity was described as “modest”, therefore it has been proposed to explore this compound in combination with other agents for future studies [136].

VS-6062 was the first FAK inhibitor tested in clinical phase I trials (NCT00666926) but, despite promising results, it was discontinued due to nonlinear pharmacokinetics [137]. Similarly, VS-4718 is also no longer in active clinical development [138]. Almost 10 years ago, Verastem developed VS-6063 (Defactinib), a second-generation compound with a more favorable pharmacokinetics. Defactinib is by far the most studied compound in the clinic. As acceptable phase I safety profiles were obtained in patients with advanced solid tumors [139,140], phase II combinatorial clinical trials with Defactinib are currently under evaluation in different advanced solid tumors. Around sixteen studies have reached phases I and II in different clinical trials, of them, six are currently in “active” or “recruiting” state. Although several attempts have been made in testing the effect of Defactinib in patients with malignant mesothelioma, due in part to the fact that merlin deficiency has been shown to predict FAK-inhibitor sensitivity [141], several trials were discontinued due to insufficient efficacy (NCT02004028, NCT01870609, and NCT02372227). Nevertheless, a phase I study evaluating Defactinib in combination with the immune checkpoint inhibitor Pembrolizumab is planned for patients with pleural malignant mesothelioma (NCT04201145), and another phase II study is currently evaluating the effects of this drug as single agents in patients with *NF2* mutations (NCT04439331). Regarding patients with KRAS mutant lung cancer (NCT01951690), Defactinib monotherapy demonstrated modest clinical activity [142]. At present time, other phase I/II studies that are in “active” or “recruiting” phases, are evaluating the effects of Defactinib in combination with Pembrolizumab (NCT03727880 and NCT02758587), Pembrolizumab, and Gemcitabine (NCT02546531), or RAF/MEK inhibitors (NCT03875820) in patients with different advanced solid tumors. A phase I study of Defactinib in combination with Paclitaxel (NCT01778803) in patients with advanced ovarian cancer exhibited signs of modest activity. A phase I/II clinical trial for recurrent platinum-resistant ovarian cancer termed ROCKIF has been initiated (NCT03287271), and will investigate the combination of Defactinib, Carboplatin and Paclitaxel.

Altogether, it seems that the FAK inhibitors examined in different clinical trials have shown manageable toxicity profiles, some efficacy in delaying progression-free survival and maintaining stable disease in different advanced solid tumors; however, they did not produce objective clinical responses. Trials are currently directed to strengthen the efficacy by combining FAK inhibitors with cytotoxic chemotherapy, targeted therapy, or immunotherapy. In the future, prognostic markers could help to select patients who could benefit from FAK inhibitor treatment alone or in combination strategies [143].

## 7. Potential Benefits of Targeting the Crosstalk Wnt–FAK

An important raising question is whether FAK inhibitors are effective as single agents. Since FAK is activated by so many binding partners and is interconnected with different signaling pathways, a combinatorial therapy seems the most promising approach [4]. A number of preclinical studies have shown that cancer cells adapt to inhibition of signaling pathway circuits by establishing alternative signaling routes through crosstalks in order to evade drug treatments [144]. Hence, it will be important not only to inhibit primary signaling pathways that induce tumorigenesis, but concomitantly, prevent pro-survival pathway crosstalks [115].

The combinatorial therapy using FAK inhibitors with agents blocking other molecules or signaling pathways has already shown effectiveness over the use of single inhibitors in the preclinical setting. In a study of acute lymphoblastic leukemia, FAK downregulation enhanced the inhibitory effect of the mTOR inhibitor Rapamycin on cell growth in vitro, slowed the progression of leukemia in NOD/SCID mice in vivo, and prolonged the median survival time of the mice [145]. Dual inhibition of FAK and EGFR signaling pathways enhanced apoptosis in breast cancers cells [146], while simultaneous inhibition of FAK and Src increased the apoptosis of colon cancer cell lines, involving the blockade of the AKT survival pathway [147]. The connection between PTEN and FAK has also emerged as a potential druggable signaling axis. Loss of PTEN causes the activation of the pro-survival pathway PI3K/AKT/mTOR, but results obtained with specific inhibitors of PI3K/AKT/mTOR are not so encouraging. As reduced PTEN activity often correlated with increased expression of FAK [148], a combinatorial strategy was proposed based on concomitant inhibition of PI3K/AKT and FAK signaling, in advanced metastatic malignancies with altered or reduced PTEN expression [149]. A recent study showed that in vitro downregulation of Calretinin, a calcium-binding protein expressed in malignant mesothelioma [150] and activator of the FAK signaling pathway [29], increases the susceptibility of malignant mesothelioma cells to the FAK inhibitor Defactinib in vitro [84]. However, surviving cells showed increased resistance to chemotherapy due to the activation of the Wnt signaling pathway, demonstrating the induction of a secondary oncogenic pathway conferring resistance to therapy.

The many existing Wnt–FAK crosstalks in different cancers are complex, variable, and cell-type dependent. There is emerging in vitro and in vivo evidence that disrupting these crosstalks at multiple levels has therapeutic benefits. Pharmacological inhibition of FAK/PYK2 repressed adenoma formation in APC^min/+^ mice and reduced phospho-GSK3β and β-catenin intestinal levels, decreasing Wnt signaling [74]. Co-targeting the Wnt pathway using FAK inhibitors was proposed as a new approach to overcome resistance to BRAF inhibitors in BRAF-mutated colorectal cancer patients [76]. Indeed, FAK inhibitors in combination with BRAF inhibitors, were shown effective in decreasing tumor growth in a mouse xenograft model [75]. The inhibition of FAK blocks tumor proliferation and prolongs the survival in a c-Met/β-catenin-driven hepatocellular carcinoma (HCC) mouse model [104], suggesting that inhibition of FAK might be beneficial for treating HCC, especially in patients showing activated β-catenin. In various malignant mesothelioma cell lines, the pharmacological inhibition of both pathways decreased dramatically cell proliferation and survival, indicating that the simultaneous targeting of Wnt and FAK might be considered as a prospective strategy for malignant mesothelioma treatment [84].

Nevertheless, despite the many examples of FAK–Wnt interactions described in the literature and summarized here, we are still far away from a comprehensive picture of this crosstalk, especially regarding its different cancer-specific dynamics. Some studies have started to develop mathematical models or network-based computational methods to quantify the signaling crosstalks in order to identify new therapeutic targets and evaluate the most effective drug target combinations [115,151].

## 8. Conclusions and Future Perspectives

In this review, we have highlighted the current knowledge on Wnt–FAK signaling crosstalks in different cancers (summarized in Figure 1). FAK appears to be an important regulator of the Wnt signaling pathway. However, we found a wide range of FAK–Wnt interactions at different levels that are context- and cell-type dependent; thus, a better understanding of the nature and physiological relevance of this interaction is necessary [56], before devising combinatorial therapeutic strategies.

Clinical trials with Wnt and FAK inhibitors have demonstrated some limitations, in part due to the important roles that both pathways exert in the maintenance of adult homeostasis. The future use of Wnt inhibitors have been restricted as a result of severe side effects; and despite current data that justify the use of FAK inhibitors in solid tumors, single-agent targeting shows still modest clinical activity. Therefore, efforts are nowadays directed in testing both inhibitors in combination with different chemotherapeutic agents and/or cell signaling/checkpoint inhibitors.

Both indirect inhibition of the FAK–Wnt crosstalk by targeting FAK (consequently reducing Wnt activity), or a combinatorial approach targeting key molecules from both pathways (thus disrupting compensatory signaling mechanisms), seem promising strategies. In fact, synergistic drug combinations are generally more specific to particular cellular contexts improving the therapeutically selectivity [156]. The proper design of drug combinations requires a profound understanding of the crosstalks between oncogenic pathways, in addition to other factors, such as intercellular signaling responses in the cancer microenvironment and drug responses in individual cells [157]. Hence, a multidisciplinary approach integrating cancer cell biology, animal and 3D models, biomarker analysis, high-throughput technologies and network-based computational methods seems more and more essential for the discovery of new effective treatments. We believe that additional FAK–Wnt crosstalks will be revealed in the future and that effective and alternative strategies will soon emerge to overcome their signaling crosstalk-mediated effects during tumorigenesis.

## Figures and Tables

**Figure 1 ijms-21-09107-f001:**
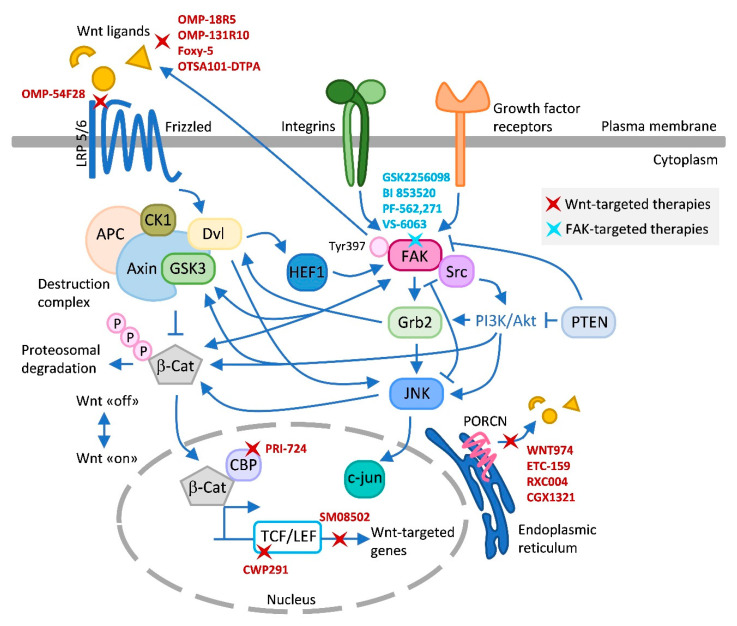
Wnt–FAK signaling crosstalks and inhibitors. This figure summarizes the main known crosstalks existing between Wnt and FAK that have been described in the text and in the literature, and the different Wnt and FAK inhibitors tested in clinical trials. Upon FAK activation, the FAK/Src complex phosphorylates and recruits several downstream signaling targets, including PI3K/AKT. GSK3 generally acts as a downstream signaling protein molecule of AKT [152]. Grb2 coordinates signaling downstream of integrin/FAK to activate JNK. Grb2 also interacts directly with Dvl [153]. Dvl can stimulate c-Jun-dependent transcription activity and the kinase activity of JNK [154]. Loss of PTEN function causes the activation of PI3K/AKT and JNK pathways [155]. PTEN also controls FAK [148]. FAK and PYK2 promote Wnt/β-catenin pathway activation by phosphorylating GSK3β [74]. This phosphorylation inhibits the activity of GSK3β which otherwise would drive rapid degradation of β-catenin. FAK increases expression of Wnt ligands activating Wnt signaling and CSC self-renewal indirectly or directly by activating β-catenin [102]. In addition, FAK was shown to trigger the β-catenin signaling pathway through nuclear translocation of β-catenin and transcriptional activation of β-catenin target genes [74]. FAK and Wnt have been described to modulate each other antagonistically [84]. HEF1 localizes to focal adhesions to coordinate FAK/Src signaling and is also modulated by Wnt [80]. Wnt-targeted agents include OMP18R5, OMP131R10, Foxy-5, OTSA101-DPTA, and OMP-54F28, which target Wnt signaling at the ligand/receptor level; PRI-724, CWP291, and SM08502 at the transcriptional level; and the Porcupine inhibitors WNT974, ETC-159, RXC004, and CGK1321, which block Wnt ligand secretion. FAK inhibitors include GSK2256098, PF-562,271, and VS-6063, which competitively target the ATP-binding site K454, located in the kinase domain of FAK; and the competitive scaffold inhibitor BI 853520, which binds to the hinge region of the kinase domain of FAK blocking the access of ATP to the ATP binding site [143]. Arrows (↑) indicate activation/induction, and blunt-ended lines (T) indicate inhibition/blockade. LRP, low-density lipoprotein receptor-related protein; Dvl, dishevelled; CK1, casein kinase 1; APC, adenomatous polyposis coli; GSK3, glycogen synthase kinase-3; β-Cat, β-catenin; CBP, CREB binding protein; TCF/LEF, T-cell factor/lymphoid enhancer factor; Grb-2, growth factor receptor-bound protein 2; PTEN, phosphatase and tensin homolog; JNK, c-Jun N-terminal kinase; PORCN, Porcupine.

**Table 1 ijms-21-09107-t001:** Summary of Wnt signaling inhibitors in clinical trials.

Name	Target/Mode of Action	Development Phase	Condition or Disease	Status	Trial Identifier
OMP-18R5 (Vantictumab)	Anti-Fzd7 antibody	Phase I	Advanced solid tumors; metastatic breast cancer; pancreatic cancer	Completed	NCT01345201 NCT01973309 NCT02005315 NCT01957007
OMP-54F28 (Ipafricept)	Fzd8 decoy receptor	Phase I	Advanced solid tumors; ovarian cancer; hepatocellular cancer; pancreatic cancer	Completed	NCT01608867 NCT02092363 NCT02069145 NCT02050178
OMP-131R10 (Rosmantuzumab)	Anti-R-spondin3 antibody	Phase I	Advanced relapsed tumors; refractory solid tumors	Completed	NCT02482441
Foxy-5	Wnt-5a mimicking peptide	Phase II	Colon cancer	Recruiting	NCT03883802
OTSA 101-DPTA	Anti Fzd10 antibody	Phase I	Relapsed or refractory synovial sarcoma	Recruiting	NCT04176016
PRI-724	Inhibitor β-catenin-CBP	Phase I/II	Advances solid tumors; chronic/acute myeloid leukemia; pancreatic cancer	Terminated or Completed	NCT01302405 NCT01606579 NCT01764477
CWP291	Sam68	Phase I	Acute/chronic myeloid leukemia	Completed	NCT01398462
SM08502	CLK	Phase I	Advanced solid tumors	Recruiting	NCT03355066
Wnt974 (LGK974)	Porcupine inhibitor	Phase I	Advanced solid tumors	Recruiting	NCT01351103
ETC-159	Porcupine inhibitor	Phase I	Advanced solid tumors	Recruiting	NCT02521844
RXC004	Porcupine inhibitor	Phase I	Advanced solid tumors	Recruiting	NCT03447470
CGX1321	Porcupine inhibitor	Phase I	Advanced GI tumors	Recruiting	NCT02675946

Fzd, frizzled; CBP, CREB binding protein; CLK, CDC-like kinase; GI, gastrointestinal.

**Table 2 ijms-21-09107-t002:** Summary of FAK signaling inhibitors in clinical trials.

Name	Target/Mode of Action	Development Phase	Condition or Disease	Status	Trial Identifier
GSK2256098	ATP-competitive kinase inhibitor	Phase II	Pancreatic cancer; adenocarcinoma	Active, not recruiting	NCT02428270
		Phase I	Mesothelioma; solid tumors	Completed	NCT01138033 NCT01938443
		Phase II	Meningioma	Suspended	NCT02523014
BI 853520	ATP-competitive kinase inhibitor	Phase I	Advanced or metastatic cancers	Completed	NCT01335269 NCT01905111
PF-562271 (VS-6062)	ATP-competitive kinase inhibitor	Phase I	Head and neck cancer; prostatic cancer; pancreatic cancer	Completed	NCT00666926
PND-1186 (VS-4718)	ATP-competitive kinase inhibitor	Phase I	Pancreatic cancer; non-hematologic or metastatic cancers; leukemia	Terminated or withdrawn	NCT02651727 NCT01849744 NCT02215629
Defactinib (VS-6063; PF-04554878)	ATP-competitive kinase inhibitor	Phase I	Malignant pleural mesothelioma	Not yet recruiting	NCT04201145
		Phase I	NSCLC; solid tumors; low grade serous ovarian cancer; colorectal cancer	Recruiting	NCT03875820
		Phase I/II	Ovarian cancer	Recruiting	NCT03287271
		Phase I/II	Carcinoma; NSCLC; mesothelioma; pancreatic cancer	Recruiting	NCT02758587
		Phase II	Pancreatic ductal adenocarcinoma	Recruiting	NCT03727880
		Phase I	Advanced solid tumors; pancreatic cancer	Active, not recruiting	NCT02546531
		Phase II	Cancers with *NF2* genetic changes (advanced lymphoma; advanced solid cancers; hematopoietic cancers, etc.)	Active, not recruiting	NCT04439331
		Phase II	Patients with *KRAS* mutations (NSCLC; lung cancer)	Completed	NCT01951690
		Phase I	Ovarian cancer	Completed	NCT01778803
		Phase I	Non-hematologic cancers	Completed	NCT01943292
		Phase I	Advanced non-hematologic malignancies	Completed	NCT00787033
		Phase I	Healthy subjects	Completed	NCT02913716
		Phase II	Malignant pleural mesothelioma	Terminated	NCT02004028
		Phase I	Epithelial ovarian cancer	Terminated	NCT02943317
		Phase II	Malignant pleural mesothelioma	Terminated	NCT01870609
		Phase I	Relapsed malignant mesothelioma	Terminated	NCT02372227

*NF2*, neurofibromatosis 2; NSCLC, non-small-cell lung cancer; *KRAS*, Kirsten rat sarcoma 2 viral oncogene homolog.

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
