# Peer review of "The Crosstalk between FAK and Wnt Signaling Pathways in Cancer and Its Therapeutic Implication"

_ijms, 2020, doi:10.3390/ijms21239107_

Round 1
Reviewer 1 Report
The authors present a comprehensive review of the crosstalk between FAK and Wnt signalling pathways in cancer. The review article is well written, thoughtfully structured and adequately referenced. There is a logical structure in which they summarise the many complex interactions between FAK and Wnt in different cancers. They authors should be commended on their efforts and I believe that this review will be of high interest to anyone working in the field of FAK, Wnt of cancer signally pathways.
I have suggested only minor typographical, grammatical corrections, which I have annotated in the attached document.
My only major concern was to ask the authors to check their statement “…pointing towards insufficient specificity of this agent [121].” (page 8, lines 360-361. Please qualify this statement by clarifying whether this conclusion is based on the absence of information from the corresponding trial. It may seem misleading, or inappropriate to form a conclusion (one way or the other), when little information is available.
Otherwise, I’m happy to recommend accepting the review for publication once minor edits are made.

Author Response
We would like to thank Reviewer 1 for the time spent on reviewing our manuscript and his/her positive and helpful comments.
Accordingly, we have revised and corrected all the minor typographical and grammatical corrections suggested by the reviewer throughout the manuscript.
Regarding the major concern about the statement “…pointing towards insufficient specificity of this agent [121]” we agree with the reviewer that this statement might be confusing, so we have revised it and now the text reads as follows: “In the case of Rosmantuzumab (OMP-131R10), an anti-R-spondin 3 antibody, there is little information about the outcome of the phase I clinical trial (Clinical trial identifier: NCT02482441), being just reported that Rosmantuzumab “failed to provide compelling evidence of clinical benefit” (new reference: OncoMed Pharmaceuticals, Inc. Annual Report [Internet]. Available from: https://sec.report/Document/1302573/000156459019006795/omed-10k_20181231.htm).
Reviewer 2 Report
The authors have prepared a great manuscript for the review on the connection between Fak signaling and Wnt for Cancer. This will be a major help to the field of cancer biology in getting a reference to getting knowledge about how to target this pathway for therapeutic applications. The section on Wnt inhibitors is fantastic and gives a clear and current perspective on the ongoing therapeutic approaches on targeting this pathway.
Author Response
We would like to thank Reviewer 2 for the time spent on reviewing our manuscript and his/her positive comments.